# Characterization of the Chloroplast Genome of *Trentepohlia odorata* (Trentepohliales, Chlorophyta), and Discussion of its Taxonomy

**DOI:** 10.3390/ijms20071774

**Published:** 2019-04-10

**Authors:** Huan Zhu, Yuxin Hu, Feng Liu, Zhengyu Hu, Guoxiang Liu

**Affiliations:** 1Key Laboratory of Algal Biology, Institute of Hydrobiology, Chinese Academy of Sciences, Wuhan 430072, China; huanzhu@ihb.ac.cn (H.Z.); ssshyx@163.com (Y.H.); 2Key Laboratory of Marine Ecology and Environmental Sciences, Institute of Oceanology, Chinese Academy of Sciences, Qingdao 266071, China; liufeng@qdio.ac.cn; 3State Key Laboratory of Freshwater Ecology and Biotechnology, Institute of Hydrobiology, Chinese Academy of Sciences, Wuhan 430072, China; huzy@ihb.ac.cn

**Keywords:** chloroplast genome, free-standing ORFs, introns, phylogenetic analysis, taxonomic study, *Trentepohlia odorata*, Trentepohliales

## Abstract

Trentepohliales is an aerial order of Chlorophyta with approximately 80 species distributed mainly in tropical and subtropical regions. The taxonomy of this genus is quite difficult and presents a challenge for many phycologists. Although plentiful molecular data is available, most of the sequences are not identified at the species level. In the present study, we described a new specimen with detailed morphological data and identified it as *Trentepohlia odorata*. A phylogenetic analysis showed *T. odorata* as a novel lineage in Trentepohliales. *T. odorata* has the closest relationship with *T. annulata*, which is expected since sporangia of both species are without stalk cell and with dorsal pore. Species with such morphological characteristics may represent deep lineages in Trentepohliales. Although an increasing number of chloroplast genomes of Ulvophyceae have been reported in recent years, the whole plastome of Trentepohliales has not yet been reported. Thus, the chloroplast genome of *Trentepohlia odorata* was reported in the present study. The whole plastome was 399,372 bp in length, with 63 predicted protein-coding genes, 31 tRNAs, and 3 rRNAs. Additionally, we annotated 95 free-standing open reading frames, of which seven were annotated with plastid origins, 16 with eukaryotic genome origins, and 33 with bacterial genome origins. Four rpo genes (*rpoA*, *rpoB*, *rpoC1*, and *rpoC2*) were annotated within ORF clusters. These four genes were fragmented into several (partial) ORFs by in-frame stop codons. Additionally, we detected a frame shift mutation in the *rpoB* gene. The phylogenetic analysis supported that Trentepohliales clustered with Dasycladales and nested into the BDT clade (Bryopsidales, Dasycladales and Trentepohliales). Our results present the first whole chloroplast genome of a species of Trentepohliales and provided new data for understanding the evolution of the chloroplast genome in Ulvophyceae.

## 1. Introduction

Algae in the order Trentepohliales are characterized by uniseriate and branched filaments. The order contains five genera and approximately 80 species that are primarily found in tropical and subtropical areas, and many of them form masses of a striking color (such as orange, yellow, and red) on tree trunks, walls, and stones [1]. Since the recorded species are completely aerial or subaerial, Trentepohliales is a special group in Ulvophyceae, of which the other groups are mainly composed of seaweeds or freshwater algae. The order can be easily distinguished from other ulvophycean algae by its specific sporangiate-lateral. Other special morphological and ultrastructural features include the absence of pyrenoid in the net-like chloroplasts, plasmodesmata in the septa, and a phragmoplast similar to Streptophyta visible during cytokinesis [2]. As the type genus of Trentepohliaceae, *Trentepohlia* Martius is the earliest known and most studied. Several studies have confirmed that *Trentepohlia* are polyphyletic groups [3,4,5,6]. There are more than 46 valid *Trentepohlia* species recorded in AlgaeBase [1]. Among all the *Trentepohlia* species, *T. odorata* (F.H. Wiggers) Wittrock is a controversial species since several phycologists considered this species synonymous with *T. umbrina* or *T. iolithus*, whereas most phycologists treated *T. odorata* as valid separate species [7,8,9,10]. Moreover, the current molecular data (i.e., 18S rDNA, *rbcL* cpDNA) available has shown that several species in *Trentepohlia* are also polyphyletic, especially several widespread and common species, i.e., *T. arborum* and *T. umbrina* [5,6]. Additionally, the morphology and phylogenetic position of *Trentepohlia odorata* has rarely been reported.

Recent studies based on genome-wide data or multigene chloroplast data revealed that the Ulvophyceae is not monophyletic and was recovered as two or more distinct clades [11,12,13]. However, there are only several different chloroplast coding genes available in the database (i.e., GenBank), and no whole chloroplast genome of Trentepohliales has yet been reported. Chloroplasts in species of Ignatiales, Ulotrichales, Oltmannsiellopsidales, and Ulvales have been reported to share a similar structure with most Viridiplantae, of which the circular plastome was composed by two inverted repeat regions (IRa and IRb) and two single copy regions (LSC and SSC) [12,14,15,16]. The sequenced chloroplast genome of the BCDT clade (Bryopsidales, Cladophorales, Dasycladales and Trentepohliales) have unique plastome structures. For example, the chloroplast genome of a species of Cladophorales was reported to consist of 34 small hairpin chromosomes and lost many genes [17]. All available chloroplast genomes of Bryopsidales lack a large inverted repeat region [18,19]. In comparison with the relatives of Trentepohiales, there is little known about the evolution and chloroplast genome of the order. Thus, it is important to sequence the entire chloroplast genome of a species within Trentepohliales to increase our knowledge about the order.

During 2010–2016, we collected lots of Trentepohliacean specimens from China, as reported in previous study [5]. In the present study, we identified one corticolous specimen as *T. odorata* based on morphological evidence. Short- and long-read high-throughput sequencing data of this isolate were obtained and assembled. The aims of our study were: (1) to present the complete chloroplast genome of *Trentepohlia odorata*, (2) to study the taxonomy of *Trentepohlia odorata*, (3) to reconstruct phylogenetic relationship between *Trentepohlia odorata* and other species.

## 2. Results

### 2.1. Morphological Observation

*Trentepohlia odorata* (F.H. Wiggers) Wittrock 1880

Description: The alga formed a dense mat over tree bark (Figure 1A,B). The thallus mainly consisted of abundant erect parts and poorly prostrate filaments. Most of the erect filaments were nearly parallel, consisting of 3–10 cells, rarely branched and with a tapering end (Figure 1C). Apical vegetative cells of erect filaments were substantially longer than basal vegetative cells (Figure 1C,D). Cells of erect filaments were approximately 7–16 μm in width and 14–21 μm in length, with a length/width ratio of about 1.1–1.9. Prostrate filaments were often branched and form compact patches, the cells of which were mostly globose to ellipsoid or with other irregular shapes (Figure 1D,E). The size of the vegetative cells was approximately 8–13 μm in width and 12–18 μm in length, with a length/width ratio about 1.0–1.5. Presumptive zoosporangia were apical, intercalary, or lateral and mainly produced on erected filaments that are globose, ellipsoid, and obovate (Figure 1C–E). Most intercalary zoosporangia were globes and smaller than apical and lateral ones. The lateral zoosporangia were often clustered and are mainly obovate with an obvious dorsal pore (Figure 1E). The intercalary zoosporangia were approximately 8–12 μm in width and 9–16 μm in length. The lateral and apical zoosporangia were approximately 10–24 μm in width and 18–28 μm in length.

This specimen was collected from a tropical botanical garden and formed abundant red growth on the surface of *Fagus longipetiolata* trunk. The morphology of specimen DZ1317 was completely consistent with primary description by Wittrock (1880) and Printz (1939) [7,10]. Despite being unable to observe the holotype of *Trentepohlia odorata* from its type locality (Fiona, Denmark), there is little morphological difference between our specimen and the holotype according to its primary description.

### 2.2. Phylogenetic Analysis

A phylogenetic analysis based on 73 sequences recovered from species in the Trentepohliales consisted of two main clades. One main clade was composed of several small clades including the *Cephaleuros* clade, *T. aurea* clade, *T. arborum*, and *Printzina lagenifera* clade. The other main clade was composed by *Phycopeltis*, *Printzina bosseae*, *T. umbrina*, *T. iolithus*, and *T. annulata* (Appendix A). According to the 18S rDNA phylogeny, *T. odorata* had the closest relationship with *T. annulata* (KM020077) and *T.* cf. *umbrina* (KX586916) (Figure 2). According to the phylogeny based upon *rbc*L matrix, *T. odorata* clustered with Trentepohliales sp. (GU549443) and formed a robust clade with *T. annulata* (MH940266) and *T. abietina* (MH940276) (Figure 3). Such topology was consistent with previous studies. All of the analyses recovered *T. odorata* as a well resolved lineage.

Our phylogenetic analysis based on chloroplast genome showed that both Trebouxiophyceae and Ulvophyceae were paraphyletic due to Chlorellaceae and the Bryopsidales, Trentepohliales, and Dasycladales clade (BDT clade) did not fall into core Trebouxiophyceae and core Ulvophyceae, respectively (Figure 4). Both maximum likelihood analysis and Bayesian analysis strongly supported that Trentepohliales clustered with Dasycladales. In the maximum likelihood analysis, the BDT clade was supported by a bootstrap value of 60; however, such topology did not occur in the Bayesian inference analysis (Appendix A). Both maximum likelihood analysis and Bayesian inference supported that Trentepohliales may have the closest relationship with Dasycladales.

### 2.3. Chloroplast Genome Analysis

We obtained 21 contigs with a total of 355,893 bp and one circular molecule with a length 399,372 bp from short-read sequencing data and long-read sequencing data, respectively. The dot plot showed a high-level congruence between the 21 contigs and one circular molecule (Appendix A), indicating that the plastome of *T. odorata* was circular and had a length of up to 399,372 bp. Our annotation results showed that the plastome possesses the typical quadripartite structure. Two inverted repeat regions were 26,700 bp and 26,778 bp, respectively. The large single copy region (LSC) was 178,629 bp, while the small single copy region (SSC) was 167,265 bp (Figure 5). The overall G + C content of the circular cpDNA was calculated to be 29.75%. The analysis revealed that the cpDNA encodes 97 genes (Table 1).

The genes were grouped into two major categories, coding genes and non-coding genes. The coding genes consisted of 63 predicted protein-coding genes, including five *atp* genes, four *chl* genes, four *pet* genes, five *psa* genes, 15 *psb* genes, seven *rpl* genes, four *rpo* genes, 11 *rps* genes, two *ycf* genes, and six other genes (*ccsA*, two *clpP*, *infA*, *rbcL*, and *tufA*). The non-coding gene category included 31 tRNAs and three rRNAs (Table 1). We annotated a total of 49 introns, of which 39 group I introns were present in eight genes (*rrl*-IRa (8), *rrl-*IRb (8), *rrs* (2), *psaA* (2), *psbA* (3), *psbC* (5), *psbD* (3), *petB* (4), and *rbcL* (2)), nine group II introns in eight genes (*rpl*2, *rps*12, *psaA* (3), *psbA*, *psbD*, *psaC*, *petB*, and *tufA*), and one unidentified type intron (*rpoB*). The IR regions contained only two of the same rRNA gene (*rrl*) and no other genes. Additionally, we detected that the *rrs* gene was not located in IR regions but had only one copy in the SSC region. The *ycf3* gene was located across LSC region and IRb region; however, there was only a partial sequence of this gene detected in the IRa region. We annotated two copies of clpP in the LSC region and SSC region, respectively. Additionally, there were 95 free-standing ORFs (length >100 aa) annotated in the intergenic region, with a total length up to 72,720 bp (18.21%) (Figure 6). Among the free-standing ORFs, seven were annotated with plastid origins (POP), 16 with eukaryotic genome origins (EOP), and 33 with bacterial genome origins (BOP) (Figure 7). Four genes (*rpoA*, *rpoB*, *rpoC1*, and *rpoC2*) were annotated within four ORF clusters (including partial ORFs). All four genes were fragmented into several ORFs by in-frame stop codons (Figure 8, red asterisks). There were three fragments annotated in *rpoC1*, three in *rpoC2*, eight as *rpoB*, and two as *rpoA*. In the *rpoB* ORF cluster, we detected a frame shift mutation (Figure 8, arrow).

## 3. Discussion

The taxonomic controversy related to *T. odorata* primarily focused on whether or not the species was synonymous with *T. umbrina* or *T. iolithus* [7,8]. The main differences between *T. iolithus* and *T. odorata* was their substratum. *Trentepohlia iolithus* was only found on exposed stones or concrete, and *T. odorata* was found on tree bark [7,9,20]. A previous study reported *T. odorata* on other substratum, such as concrete, which was not consistent with the original description [21]. Although *T. odorata* has a very similar vegetative morphology with *T. iolithus* var. *yajiagengensis*, the morphology of sporangia and the phylogenetic position suggests that they are different species [22]. *Trentepohlia umbrina* is a paraphyletic species as sequences from *T. umbrina* clustered into several small clades in many studies. There was an obvious morphological difference between the two species. According to the original description, the thallus of *T. odorata* is heterotrichous and the thallus of *T. umbrina* is prostrate. The vegetative cells of *T. odorata* have a greater length/width ratio than that of *T. umbrina*. In the present study, our observation was consistent with the original description and the Printz description, thus we support Printz in that *T. odorata* is a morphologically distinct species, rather than Hariot [7,10]. The phylogenetic result shows that *Trentepohlia odorata* has the closest relationship with *Trentepohlia annulata*. One possible explanation for their close relationship is that both algae seem not to possess sporangiate-lateral. *Trentepohlia* species with lateral or intercalary sporangia and dorsal pore sporangia may represent several deep lineages in Trentepohliales. Additionally, there are few images regarding *Trentepohlia odorata* in previous studies, and our study provided new morphological evidence to compare those *Trentepohlia* species.

Although a considerable number of published plastomes are available, there are many gaps in Chlorophytes plastomes, especially in several orders of Ulvophyceae [12]. A recent study reported that chloroplast genomes in Cladophorales are fragmented into many small hairpin chromosomes [17]. The chloroplast genome in Bryopsidales are circular but lack a large inverted repeat [18]. Our study reported the first whole plastome of Trentepohliales, with a size up to 399,372 bp, which is the largest currently identified within Ulvophyceae. The plastome of *Trentepohlia odorata* presented a quadripartite structure, which differs from its close relatives, Cladophorales and Bryopsidales. We found several free-standing ORFs of bacterial origin and fragmentation by in-frame stop-codons in *rpoA*, *rpoB*, *rpoC1*, and *rpoC2* genes, which is similar with Bryopsidales [18]. The *rrf* gene was not detected in this study. We detected that the *rrl* gene located in IR region, and the *rrs* gene located in the SSC region. Fragmentation by introns were found in the *rrl* gene, with eight group-I introns. Similar cases were also found in *Caulerpa manorensis*, *Jenufa perforata*, *Schizomeris leibleinii*, and *Floydiella terrestris*, with seven, five, seven, and eight introns, respectively [18,19,23,24]. Two copies of the *clpP* gene were annotated. This gene duplication is very common in the nuclear genome and is usually caused by two repeats located at the two sides of the genes in the organellar genome. However, we did not detect repeat sequences at the sides of the two *clpP* genes. Our phylogenetic analysis using chloroplast genomes indicated that both Ulvophyceae and Trebouxiophyceae are paraphyletic, which is consistent with previous studies [11,13]. Trentepohliales clustered with Dasycladales in present study, which is also reported by previous studies [11,13]. However, we cannot rule out that Trentepohliales have a closest relationship with Cladophorales since species in Cladophorales was not included in our phylogenomic analysis.

## 4. Materials and Methods

### 4.1. Morphology and Cultivation

The specimen of *T. odorata* was derived from samples collected from the tree bark of *Fagus longipetiolata* located in the Hainan Tropical Botany Garden (19°30′49″ N, 109°30′14″ E) in December 2013, with the voucher number DZ1317. The specimen was primarily examined under a stereoscope microscope (Zeiss model KL1500 LCD; Carl Zeiss, Göttingen, Germany). Filaments were isolated using forceps and dissecting needles. We selected clean and complete filaments, placed them in culture dishes, and then spread the branches while using a stereoscope. Morphology was observed under differential interference contrast microscopy using a Leica DM5000B microscope (Leica Microsystems, Wetzlar, Germany). Micrographs were captured using a Leica DFC320 digital camera. Voucher specimens were deposited in the Freshwater Algal Herbarium (IHB), Institute of Hydrobiology, Chinese Academy of Sciences. The unialgal strain was aerially cultivated in a photo-reactor described by Chen et al. [25].

### 4.2. DNA Extraction, PCR, and Phylogenetic Reconstruction

The genomic DNA was extracted using a Axygen Universal DNA Isolation Kit (Axygen, Suzhou, China). Partial 18S rDNA was obtained as described in Zhu et al. (2017). The PCR products were purified and then sent to Tsingke Biotech Company, Inc. (Wuhan, China) for sequencing. Additional 18S rDNA and *rbcL* sequences from Trentepohliales species were downloaded from GenBank for analyses. Sequence matrices for phylogenetic analysis were initially aligned with MAFFT 7.0 and refined manually using Seaview [26,27]. ModelFinder was utilized to select the best-fitting evolutionary models for each marker according to Bayesian information criterion calculations [28]. IQ-TREE and MrBayes3.2 were used to infer the phylogeny [29,30]. We performed two phylogenetic analyses using the 18S rDNA matrix. First, Cladophorales was used as the outgroup to infer the topology of Trentepohliales. Based upon the results of the first phylogenetic analysis, we selected *Cephaleuros* as the outgroup to perform the second analysis. In the phylogenetic analysis using the *rbc*L matrix, Ulotricales was selected as the outgroup.

### 4.3. Chloroplast Genome Assembly and Annotation

A paired-end Illumina sequencing library was prepared from total DNA using the NEBNext Ultra DNA Library Prep Kit (E7370S). The libraries were sequenced using an Illumina NovaSeq 6000 with 150 bp insertion fragments (Illumina, San Diego, CA, USA). High-throughput sequencing data was sequentially analyzed by SOAPnuke v1.3.0 and SPAdes v3.10.0 [31,32]. A 1D genomic DNA by ligation (SQK-LSK108) kit was used to construct a long-reads library according to the manufacturer’s instructions. The prepared library was loaded on Oxford Nanopore GridION X5 platform and sequenced. Only reads with mean scores >7 were retained. Long-read sequencing data were assembled using Canu v.1.6.0 [33]. The contigs were used to screen the chloroplast genome using the Blast program [34]. The selected chloroplast genome contig was assembled using Sequencher 4.10. Following this, we used Geneious 8.1 to map all the reads to the spliced genome sequence to verify that the contig was concatenated [35]. Finally, we obtained 21 chloroplast contigs from short-read sequencing data and one circular molecule from long-read sequencing data. A multicollinear dot plot was performed using Genome Pair Rapid Dotter v1.40 to detect the homology between 21 contigs and one circular molecule [36].

The chloroplast genome was primarily annotated using the online program DOGMA (Wyman et al., 2004) [37] and MAKER [38]. All open reading frames (ORFs) (with length >300 bp) were extracted by ORFfinder (https://www.ncbi.nlm.nih.gov/orffinder/), and then BLASTn and BLASTp (http://blast.ncbi.nlm.nih.gov/) were used with the e value set to 1e-10 to annotate the free-standing ORFs. Transfer RNA and ribosomal RNA genes were identified using tRNAscan-SE v1.23 and RNAmmer, respectively [39,40]. Intron boundaries were determined by modeling intron secondary structures and by comparing intron-containing genes with intronless homologs [41,42]. The graphical gene map was designed with Organellar Genome DRAW program (https://chlorobox.mpimp-golm.mpg.de/OGDraw.html) [43]. The annotated chloroplast genome was submitted to GenBank under the accession number MK580484.

We obtained a nucleoid dataset of 16,359 unambiguously aligned positions consisting of 31 common cpDNA-encoded genes of 43 Chlorophytes from Genbank (https://www.ncbi.nlm.nih.gov/genbank/), of which *Picocystis salinarum* and *Nephroselmis astigmatica* were used as the outgroup taxa. Most the genome accession numbers were presented before species name in Figure 8. Accession numbers of *Cephaleuros* sp., *Acetabularia peniculus* and *Scotinosphaera* sp. chloroplast genome sequences were MG721699-MG721754, MH545187-MH545222, and MG721898-MG721961 respectively. The data partition and best-fit models were selected using ModelFinder according to Bayesian inference criteria [28]. We used IQtree v1.7 and MrBayes 3.2 to perform maximum-likelihood analysis and Bayesian inference, respectively [29,30]. Additionally, because genes in the Cladophorales plastome were unique and there were fewer genes than the chloroplast genomes of other green algae, we did not include it in our analysis.

## 5. Conclusions

The morphological observation and phylogenetic analysis recovered *Trentepohlia odorata* as a separate species. According to phylogenetic results, *Trentepohlia odorata* has the closest relationship with *Trentepohlia annulata*, which could be explained by the shared morphological evidence, i.e., the absent sporangiate-lateral. The chloroplast genome of *Trentepohlia odorata* is 399,372 bp, with 97 genes and 95 free-standing ORFs. The typical quadripartite structure of *Trentepohlia odorata* is different from Cladophorales and Bryopsidales. However, the fragmention of *rpo* gene clusters by in-frame stop codons and the frame shift mutation detected in the present study is similar to Bryopsidales. Our study describes the first chloroplast genome of Trentepohliales and provides new data for understanding the evolution of the chloroplast genome in BCDT clade. Since Trentepohliles is an exclusively aerial order in core Chlorophytes, future study focused on plastomes of the other Trentepohliaceaen groups such as *Trentepohlia aurea*, *Trentepohlia bosseae* and *Cephaleuros* might have more interesting findings.

## Figures and Tables

**Figure 1 ijms-20-01774-f001:**
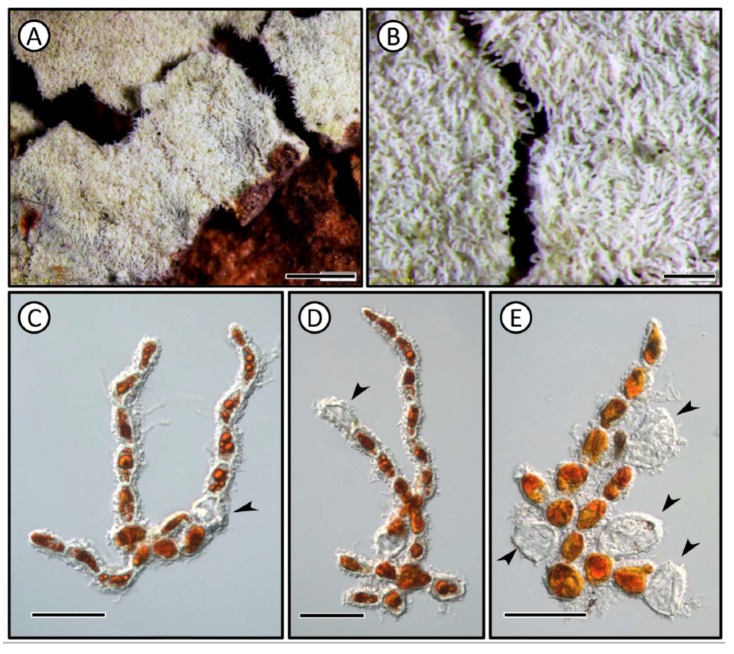
Morphology of *Trentepohlia odorata*. (**A**,**B**), steroscopic view of *Trentepohlia odorata* specimen (preserved in Formol acetic alcohol solution) on surface of *Fagus longipetiolata*. (**C**), microscopic view of heterotrichous thallus and its intercalary zoosporangium (arrow). (**D**), the apical zoosporangium with a dorsal pore (arrow). (**E**), the lateral zoosporangia with dorsal pore in cluster (arrows) at the basal part of thallus. Scale bars, 1 mm in (**A**), 200 μm in (**B**), 30 μm in (**C**–**E**).

**Figure 2 ijms-20-01774-f002:**
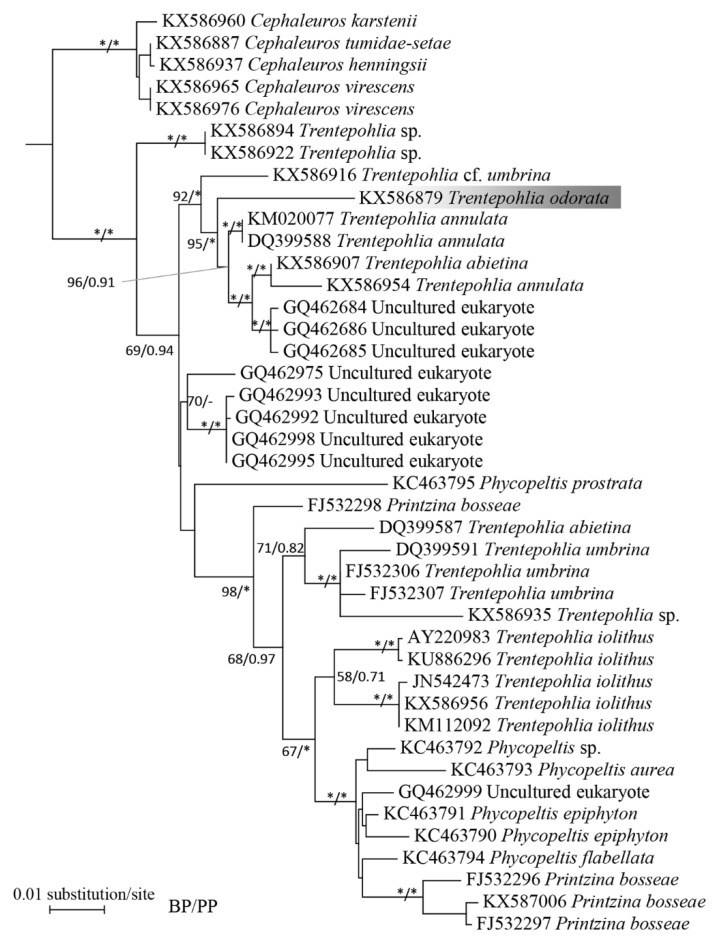
The maximum likelihood phylogram inferred from 18S rDNA sequences of Trentepohliales. The monophyletic genus *Cephaleuros* was used as outgroup. Maximum likelihood bootstrap values (1000 replicates) and Bayesian posterior probabilities (PPs) are given near the nodes, and only bootstrap proportion (BP) and PP values above 50 and 0.50 are shown. The full statistical support (100/1.00) is marked with an asterisk. The *Trentepohlia odorata* was shaded in grey.

**Figure 3 ijms-20-01774-f003:**
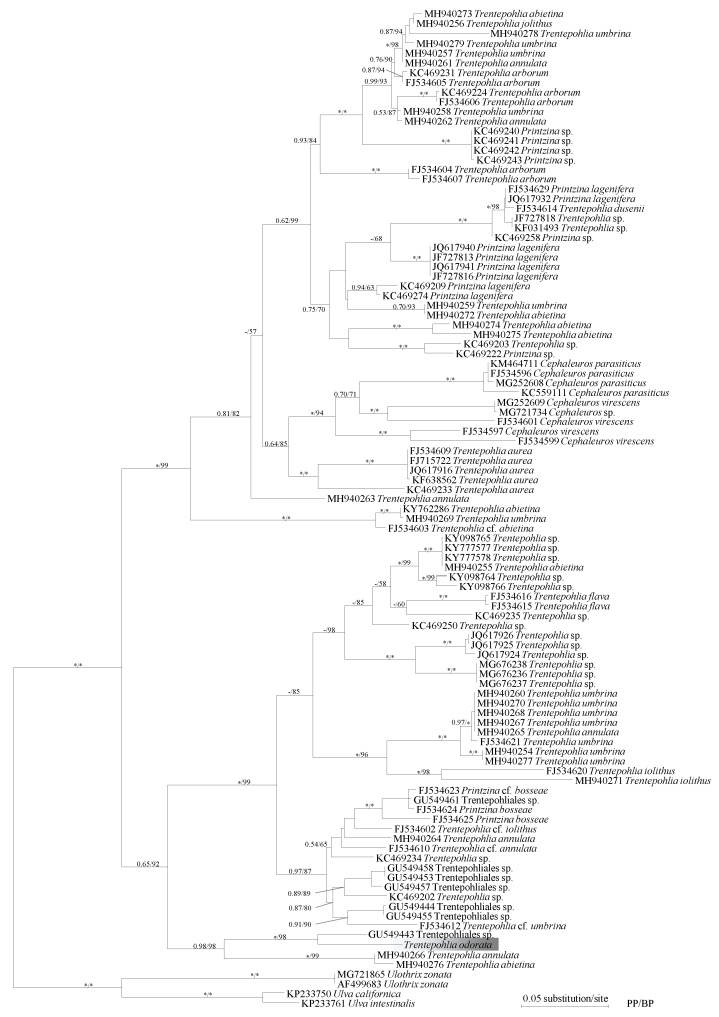
Maximum likelihood tree of Trentepohliales based on rbcL sequence data. Bayesian posterior probability (pp ≥ 0.50) and maximum likelihood (ML ≥ 50) bootstrap values are shown near the branches. The asterisks represent full statistical support (1.00/100). The *Trentepohlia odorata* was shaded in grey.

**Figure 4 ijms-20-01774-f004:**
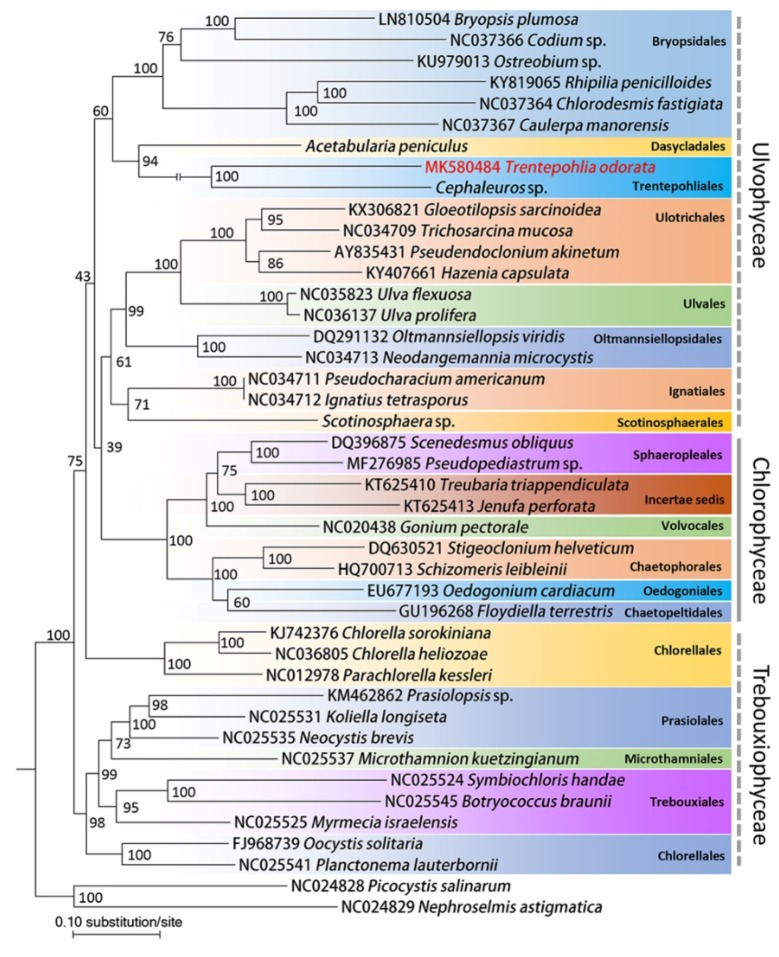
Maximum likelihood tree inferred from a dataset consisted of 31 common cpDNA-encoded genes of 43 core Chlorophytes. The genera *Picocystis* (NC024828) and *Nephroselmis* (NC024829) were selected as outgroup. Maximum likelihood bootstrap values (1000 replicates) are given near the nodes. Taxonomic arrangement follows AlgaeBase, and *Trentepohlia odorata* is in red.

**Figure 5 ijms-20-01774-f005:**
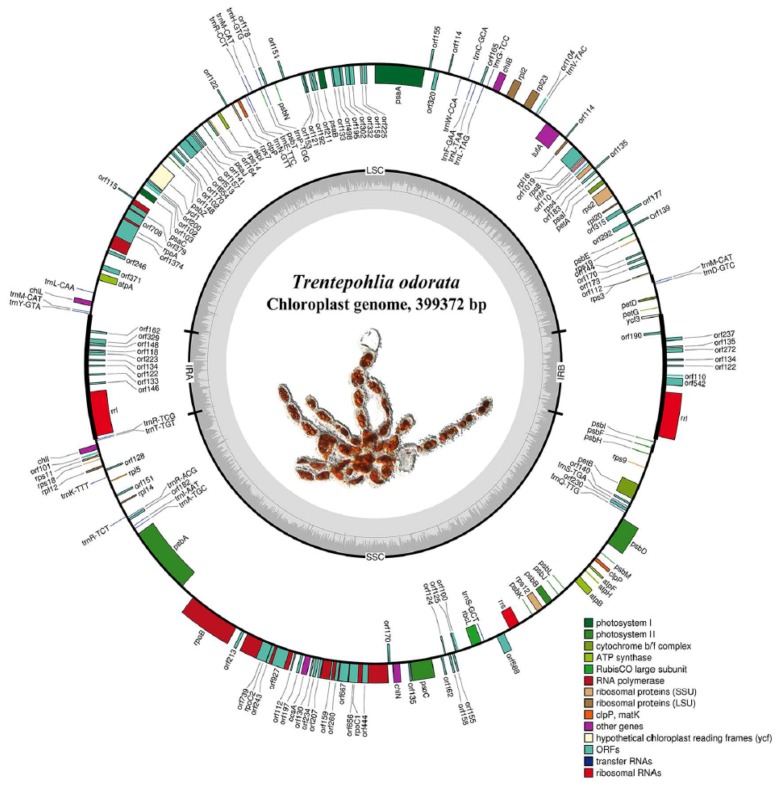
Circular map of the chloroplast genome of the Trentepohlia odorata (MK580484). Genes are color coded according to the functional categories listed in the index below the map. The GC content and inverted repeats (IRA and IRB) which separate the genome into two single copy regions are indicated on the inner circle. Genes on the inside of the outside circle are transcribed in a clockwise direction; those on the outside of the map are transcribed counterclockwise.

**Figure 6 ijms-20-01774-f006:**
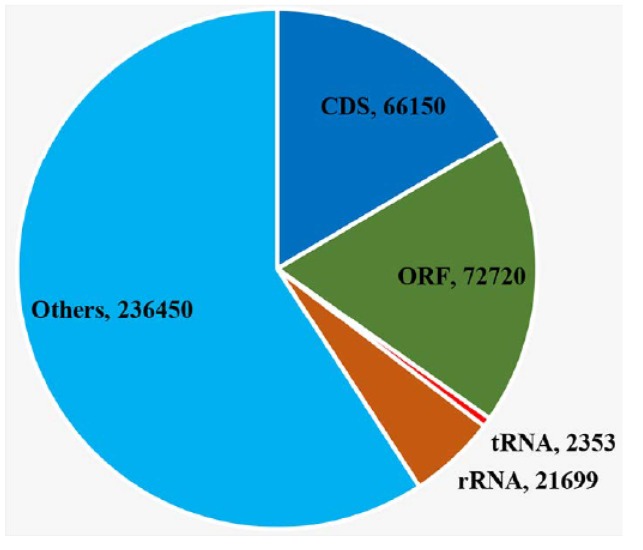
The size of CDS (protein coding regions), ORF (open reading frames, >100 aa), tRNA, rRNA and other regions in *Trentepohlia odorata* chloroplast genome. The number represents its size (bp).

**Figure 7 ijms-20-01774-f007:**
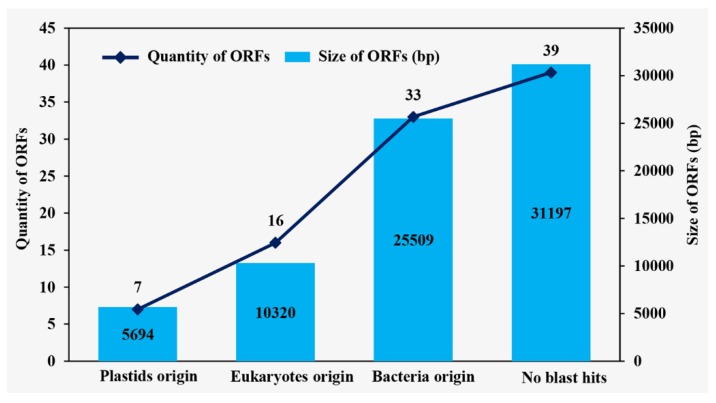
Blast result of free-standing ORFs in *Trentepohlia odorata* chloroplast genome.

**Figure 8 ijms-20-01774-f008:**
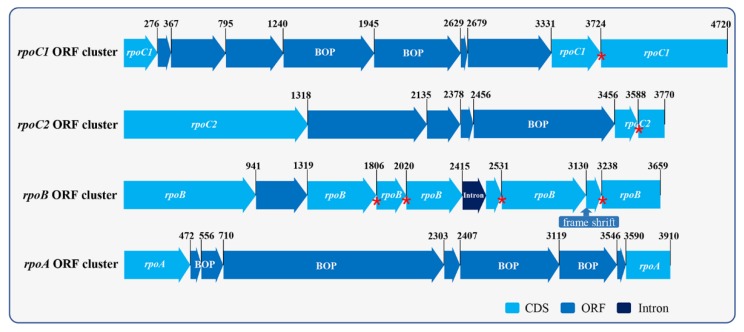
The fragmentation of *rpo* gene cluster. The red asterisk represents a detected in-frame stop codon; the shift mutation was labelled in a blue arrow. The BOP, EOP and POP represent ORF may have a bacteria, eukaryotic nucleus, and chloroplast origin respectively. ORF without BOP, EOP or POP label means no blast hit.

**Table 1 ijms-20-01774-t001:** Genes encoded by *Trentepohlia odorata* chloroplast genome.

Gene Products	Genes
ATP synthase	*atp*A, B, F, H, I
Chlorophyll biosynthesis	*chl*B, I, L, N
Cytochrome b6/f	*pet*A, B, D, G
Photosystem I	*psa*A, B, C, I, J
Photosystem II	*psb*A, B, C, D, E, F, H, I, J, K, L, M, N, T, Z
Large subunit ribosomal proteins	*rpl*2, 5, 12, 14, 16, 20, 23
RNA polymerase	*rpo*A, B, C1, C2
Small subunit ribosomal proteins	*rps*2, 3, 4, 7, 8, 9, 11, 12, 14, 18, 19
Unknown function proteins	*ycf*1, 3
Other proteins	*clp*p (2 copies), *ccs*A
Rubisco	*rbc*L
Translation factors	*tuf*A, *inf*A
Ribosomal RNAs	*rrl* (2 copies), *rrs*
Transfer RNAs	*trn*A(TGC), C(GCA), D(GTC), E(TTC), F(GAA), G(TCC), H(GTG), I(AAT), K(TTT), L(CAA), L(TAA), L(TAG), M(CAT) (5 copies), N(GTT), P(TGG), Q(TTG), R(ACG), R(CCT), R(TCG), R(TCT), S(GCT), S(TGA), T(TGT), V(TAC), W(CCA), Y(GTA)

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
