# Peer review of "Characterization of the Chloroplast Genome of Trentepohlia odorata (Trentepohliales, Chlorophyta), and Discussion of its Taxonomy"

_ijms, 2019, doi:10.3390/ijms20071774_

Reviewer 1 Report

In general, this is a well-written manuscript. Concerning, I think, an important taxonomic issue. There are several issues that in my opinion require clarification, additions or reflections.

Certainly I would consider changing the title here, there are not too many taxonomic discussions. Rather, it is a descriptive work about the geometry characteristics of CP.

It is not entirely clear to me why the authors separated phylogenetic analyzes into two subsections. In one, they analyze 18S rDNA and rbcL and in the second section the whole genome. My considerable concerns are also raised by the variable number / variable species composition in each of the above analyzes. It is obvious that by analyzing different species compositions of different regions, we can expect different results. Question whether such an effect / confusion Authors wanted to get deliberately? It seems to me that analyzes should be carried out on a fixed species composition and possibly a variable set of markers. How to compare such results obtained for various species compositions?

In my opinion, change requires either the title or content of the abstract. Because at the moment there is nothing about the taxonomic discussion in the abstraction, there is a lot about the genome's characteristics. In general, this title is not very scientific.

In MS there are common errors in the spelling of the Trentepohlia odorata species.

There are also not too lucky wordings or illogical and non-stylistic sentences.

I also miss some conclusions and future perspectives.

Author Response

In general, this is a well-written manuscript. Concerning, I think, an important taxonomic issue. There are several issues that in my opinion require clarification, additions or reflections.

Certainly I would consider changing the title here, there are not too many taxonomic discussions. Rather, it is a descriptive work about the geometry characteristics of CP.

It is not entirely clear to me why the authors separated phylogenetic analyzes into two subsections. (1) In one, they analyze 18S rDNA and rbcL and in the second section the whole genome. My considerable concerns are also raised by the variable number / variable species composition in each of the above analyzes. It is obvious that by analyzing different species compositions of different regions, we can expect different results. Question whether such an effect / confusion Authors wanted to get deliberately? It seems to me that analyzes should be carried out on a fixed species composition and possibly a variable set of markers. How to compare such results obtained for various species compositions?

Re: thank you for your comments. Most errors have been fixed in our revised manuscript. All three phylogenetic analysis have put together. For detail changes, please find below. Here, I want to explain why we performed two analyses (18S rDNA and rbcL cpDNA).

(1) There are more than 40 species recorded in Algaebase, however, sequences with specie name (we considered these sequences were identified correctly) are mainly focused on several common species (i.e. T. aurea, T. arborum, T. lagenifera, T.bosseae, T. iolithus, T. abietina, T. annulata, T. umbrina, T rigidula et al. ). All those species were included in our two phylogenetic analysis. Results of our analysis once again proved that most of them are polyphyletic, which consistent with previous studies. Additionally, the phylogenetic topology of 18S rDNA and rbcL cpDNA is more or less different, which has been proved in previous studies. We agree that our analysis should include sequences or species that represent each clade either in 18S rDNA or rbcL cpDNA analysis. Thus, the phylogenetic position of Trentepohlia odorata was convincing. Besides, many species under same voucher specimen or isolates do not have 18S rDNA and rbcL cpDNA together.

(2) In my opinion, change requires either the title or content of the abstract. Because at the moment there is nothing about the taxonomic discussion in the abstraction, there is a lot about the genome's characteristics. In general, this title is not very scientific.

Re: thank you for your comment. We have added several taxonomic sentences in abstract to fit the manuscript tittle.

In MS there are common errors in the spelling of the Trentepohlia odorata species.There are also not too lucky wordings or illogical and non-stylistic sentences.

I also miss some conclusions and future perspectives.

Re: thank you for your comment. We have added conclusion part. L226-238.

Other comments:

(1)    L12. The content of the abstract does not result in any implications or taxonomic discussions. In the abstract, only the characteristics of the cp genome are described with details. In my opinion, either the title of the publication should be changed or the abstract adapted to the current title.

Re: we have changed the abstract to match the manuscript tittle.

(2)    L61. Where are the research goals? What are the research hypotheses?

Re: we have rephrased these sentences.

L66-71. “In order to study the taxonomy of Trentepohlia odorata and the chloroplast genome of Trentepohlia, we collected lots of Trentepohliacean specimens from China reported in previous study [5]. Fortunately, we identified one corticolous specimen as T. odorata based on morphological evidence. Our molecular phylogenetic analysis also recovered T. odorata as a separate species. Short- and long-read high-throughput sequencing data of this isolate were obtained and assembled. Finally, we described the whole chloroplast genome of T. odorata.”

(3)    L90. Please, all phylogenetic analyzes put together. And not to fragment into two subsections i.e. with rbcL or 18S rDNA and another one with whole genome.

Re: we have placed all three phylogenetic results together.

(4)    L102. Phylogram inferred from 18S rDNA ok. But phylogram of what?

Re: L122. Fixed. “Phylogram inferred from 18S rDNA based on Trentepohliales”.

(5)    L143. Are you sure that those results are in Figure 6?

Re: L167-170. fixed. we have changed the figure order.

(6)    L147. Please, all phylogenetic analyzes put together. And not to fragment into two subsections.

Re: fixed.

(7)    Figure 5. What is the unit on the scale / axis? Number of what? What is the unit on the scale / axis?

Re: figure 7 in revised manuscript. we have placed the units of scale /axis in the figure, please check.

(8)    Figure 6. Please, explain the meaning of placing this chart and the data contained therein. Is this data consistent with the description in the text? What does it mean daughter part?

        Re: figure 6 in revised manuscript. we have rephrased the figure legend. “Figure 6. The size of CDS (protein coding regions), ORF (open reading     frames, >100 aa), tRNA, rRNA and other regions in Trentepohlia odorata chloroplast genome. The number represent its size (bp).”

(9)    L199. What did the Authors mean? The sentence is incorrect and illogical. Are there any conclusions that can be drawn from the results and resulting from this work?

Re: L215-L230 in revised manuscript. we have rephrased this sentence, and add a conclusion part.

Reviewer 2 Report

The manuscript “Taxonomic discussion of Trentepohlia odorata (Trentepohliaceae, Chlorophyta), and description of its chloroplast genome” is well written and interesting to read.

Besides the description of the whole chloroplast genome of this species (for the first time a species of the Order Trentepohliales), this study includes a discussion about some taxonomical issues concerning this species and phylogenetic insights.

The main shortcoming, when reading the manuscript, comes from some difficulty to follow the phylogenetic discussion without a supporting explanation about the taxonomical framework of the studied groups (see example below, Ln. 148 – 154).

Some minor comments:

Abstract

Ln. 27 

‘BDT clade’ – what BDT means? I suppose that it is explained in line 149 (Bryopsidales, Trentepohliales, and Dasycladales clade) but it will be useful to refer also here.

Introduction

Ln. 46 Trentepohlia iolithus or Trentepohlia jolithus?

Ln. 47 ‘species[7‐10].’ Replace by ‘species [7‐10].’

Ln. 56 BCDT clade – explain what BCDT means

Ln. 64 ‘Additionally, the aerial reactors were used to obtain its large unialgal mass.’ – what are aerial reactors? If reference [20] is mentioned here (presently in line 212), this will become clearer.

Ln. 62-66 The way this paragraph is written looks like a synthesis of the conclusions, and not as the aims to be accomplished within this study. Perhaps rephrase.

Results

Ln. 68 - 83 The presented morphological description is the description of the specimen seen and collected in Hainan Tropical Botany Garden or pretends to be an improved description of the species?  

Was there any attempt to see if this description (based in Hainan specimen?) also matches the specimens from the type-locality (Fiona, Denmark)?

The italic font in some species names (Trentepohlia odorata, several times, and Fagus longipetiolata) is missing in legends of Figures 1, 2, 3, 4, 5 and 6.

Ln. 94 Figure S1. - It is no clear if this analysis includes Trentepohlia odorata (it seems not); the names are too small (almost illegible).

Wouldn’t it be possible to make a consensus tree from combined analysis of 18S rDNA and rbcL?

Ln. 122 Table 1.  V(Zhang et al.) – bibliographic reference is needed? It is not included in references.

Ln. 131 ‘five pet genes’ – or ‘four pet genes’?

Ln. 141 and 142 ‘freestanding ORFs’ or ‘free-standing ORFs’ – use always the same spelling.

Ln. 147 ‘In the rpoB ORF cluster, we detected a frame shift mutation (Figure 7, arrow).’ – Improve legibility; the arrow is very difficult to see.

Ln. 148 – 154 In the interpretation of phylogenetic trees (here, and also in other parts of the manuscript) the discussion about the taxonomic relationships is difficult to follow without having in mind the taxonomical framework of the groups and/or genus included in the analysis (classes, orders). Also, it is difficult to understand which the differences are between the considered taxonomic arrangement (following AlgaeBase?) and that resulting from present study. A Table, as supplementary file, with a synthesis of the taxonomy could be helpful.

Discussion

Ln. 178 – 180 Printz – refer respective work [10] Hariot – refer respective work [7]

Ln. 186 Trentepohlia odorata – in italics

Author Respons

The manuscript “Taxonomic discussion of Trentepohlia odorata (Trentepohliaceae, Chlorophyta), and description of its chloroplast genome” is well written and interesting to read.

Besides the description of the whole chloroplast genome of this species (for the first time a species of the Order Trentepohliales), this study includes a discussion about some taxonomical issues concerning this species and phylogenetic insights.

The main shortcoming, when reading the manuscript, comes from some difficulty to follow the phylogenetic discussion without a supporting explanation about the taxonomical framework of the studied groups (see example below, Ln. 148 – 154).

Re: we have added a new tree figure. In our new figure 4, the taxonomic frame of core chlorophytes would be easily read.

Some minor comments:

(1)    Ln. 27. ‘BDT clade’ – what BDT means? I suppose that it is explained in line 149 (Bryopsidales, Trentepohliales, and Dasycladales clade) but it will be useful to refer also here.

Re: fixed.

(2)    Ln. 46. Trentepohlia iolithus or Trentepohlia jolithus?

Re: fixed. Trentepohlia iolithus.

(3)    Ln. 47 species[710]. Replace by species [710].

Re: fixed.

(4)    Ln. 56 BCDT clade – explain what BCDT means

Re: fixed.

(5)    Ln. 64  ‘Additionally, the aerial reactors were used to obtain its large unialgal mass.’ – what are aerial reactors? If reference [20] is mentioned here (presently in line 212), this will become clearer.

Re: we have deleted this sentence since it will be detailed described in material and methods part.

(6)    Ln. 62-66  The way this paragraph is written looks like a synthesis of the conclusions, and not as the aims to be accomplished within this study. Perhaps rephrase.

Re: thank you for you comment. We have rephrased this paragraph. L68-73 in revised manuscript. “In order to study the taxonomy of Trentepohlia odorata and the chloroplast genome of Trentepohlia, we collected lots of Trentepohliacean specimens from China reported in previous study [5]. Fortunately, we identified one corticolous specimen as T. odorata based on morphological evidence. Our molecular phylogenetic analysis also recovered T. odorata as a separate species. Short- and long-read high-throughput sequencing data of this isolate were obtained and assembled. Finally, we described the whole chloroplast genome of T. odorata.”

Results

(7)    Ln. 68 – 83   The presented morphological description is the description of the specimen seen and collected in Hainan Tropical Botany Garden or pretends to be an improved description of the species? Was there any attempt to see if this description (based in Hainan specimen?) also matches the specimens from the type-locality (Fiona, Denmark)?

Re: thank you for our comment. we have added two sentence to discuss this question. L91-L98 in our revised manuscript. “This specimen was collected from tropical botanical garden, formed abundant red growth on the surface of Fagus longipetiolata trunk. The morphology of specimen DZ1317 was completely consisted with primary description by Wittrock (1880) and Printz (1939) [7, 10]. Despite we could not observe the holotype of Trentepohlia odorata from its type locality (Fiona, Denmark), there is little morphological difference between our specimen and the holotype according to its primary description. Maybe the main difference between two specimens was their localities and host trees.” L202-204 in revised manuscript. “Additionally, there is little pictures about Trentepohlia odorata in previous studies, and our study provided new morphological evidence to compare those Trentepohlia species.”

(8)    The italic font in some species names (Trentepohlia odorata, several times, and Fagus longipetiolata) is missing in legends of Figures 1, 2, 3, 4, 5 and 6.

Re: fixed.

(9)    Ln. 94  Figure S1. - It is no clear if this analysis includes Trentepohlia odorata (it seems not); the names are too small (almost illegible).

Wouldn’t it be possible to make a consensus tree from combined analysis of 18S rDNA and rbcL?

Re: There are more than 40 species recorded in Algaebase, however, sequences with specie name (we considered these sequences were identified correctly) are mainly focused on several common species (i.e. T. aurea, T. arborum, T. lagenifera, T.bosseae, T. iolithus, T. abietina, T. annulata, T. umbrina, T rigidula et al. ). All those species were included in our two phylogenetic analysis. Results of our analysis once again proved that most of them are polyphyletic, which consistent with previous studies. Additionally, the phylogenetic topology of 18S rDNA and rbcL cpDNA is more or less different, which has been proved in previous studies. We agree that our analysis should include sequences or species that represent each clade either in 18S rDNA or rbcL cpDNA analysis. Thus, the phylogenetic position of Trentepohlia odorata was convincing. Besides, many species under same voucher specimen or isolates do not have 18S rDNA and rbcL cpDNA together.

(10) Ln. 122  Table 1. V(Zhang et al.) – bibliographic reference is needed? It is not included in references.

Re: this just an error, and we have fixed this mistake.

(11) Ln. 131 ‘five pet genes’ – or ‘four pet genes’?

Re: fixed. four pet genes.

(12) Ln. 141 and 142 ‘freestanding ORFs’ or ‘free-standing ORFs’ – use always the same spelling.

Re: fixed. free-standing ORFs.

(13) Ln. 147 ‘In the rpoB ORF cluster, we detected a frame shift mutation (Figure 7, arrow).’ – Improve legibility; the arrow is very difficult to see.

Re: the figure has been reedited. Such mistakes have been corrected.

(14) Ln. 148 – 154    In the interpretation of phylogenetic trees (here, and also in other parts of the manuscript) the discussion about the taxonomic relationships is difficult to follow without having in mind the taxonomical framework of the groups and/or genus included in the analysis (classes, orders). Also, it is difficult to understand which the differences are between the considered taxonomic arrangement (following AlgaeBase?) and that resulting from present study. A Table, as supplementary file, with a synthesis of the taxonomy could be helpful.

Re: we have added a new tree figure. In our new figure 4, the taxonomic frame of core chlorophytes would be easily read. All the taxonomic arrangement was following AlgaeBase.

Discussion

(15) Ln. 178 – 180   Printz – refer respective work [10]; Hariot – refer respective work [7]

Re: fixed.

(16) Ln. 186   Trentepohlia odorata – in italics

Re: fixed.

Round  2

Reviewer 1 Report

This version of MS is much better. However, the title still remained the same. I still think that it should be different because it is still more about genome characteristics than about taxonomic discussions (at most the voice).

Author Response

1.  This version of MS is much better. However, the title still remained the same. I still think that it should be different because it is still more about genome characteristics than about taxonomic discussions (at most the voice).

Re: thank you for your comment. we have retitled the manuscript according to your suggestion. “Characterization of chloroplast genome of Trentepohlia odorata (Trentepohliales, Chlorophyta), and discussion of its taxonomy”.

Reviewer 2 Report

Many thanks for the revisions. I appreciate the new Figure 4 very much.

Only a few minor suggestions and comments.

Ln 46 Trentepohlia - in italic

Ln 50 Did you see the note in http://www.algaebase.org/search/species/detail/?species_id=34779 about Trentepohlia jolithus? (The name is currently applied to Trentepohlia iolithus (Linnaeus) Wallroth and there is no reason to suppose that it is incorrect (Ross & Irvine, 1967).

At least, follow always the same criteria: in Fig. 2 both names (T. iolithus and T. jolithus) are used).

Ln. 62-66 In my opinion, the objectives of the work should be presented clearly and in a more adequate format and without mentioning results. Something like: The aims of our study were: 1) to study the taxonomy of … 2) to present the complete chloroplast genome of … and 3) to reconstruct phylogenetic relationships ....

Ln. 98 I think that the sentence ‘Maybe the main difference between two specimens was their localities and host trees.’ should be omitted. This reports to a difference in habitat and not to differences in specimen’s morphology.  Besides, this species is common in Fagus species.

Ln. 139 Taxonomic arrangement was following AlgaeBase’ – replace by ‘Taxonomic arrangement follows AlgaeBase’

Ln. 200-204 Please check the English in these lines. It is hard to understand.

Ln. 203 – ‘little pictures’ or ‘few images’?

Ln. 224 could not’ or ‘cannot’?

Ln 225 was not’ or ‘were not’; ‘in in’ (repeated)

Ln. 228-239 Please check English.

Ln. 228-229 ‘… recovered Trentepohlia odorata is a separate species.’ Replace by ‘… recovered Trentepohlia odorata as a separate species.’

Ln. 237 Since Trentepohliles is ‘absolutely aerial’ – perhaps ‘exclusively aerial’ is more correct.

Ln 238 - Trentepohliceaen or Trentepohliacean?

Ln 239 I’m not understanding why Trentepohlia aurea, Trentepohlia bosseae and Cephaleuros are particularly relevant. Please explain.

Author Response

1. Ln 46. Trentepohlia - in italic

Re: fixed.

2. Ln 50. Did you see the note in http://www.algaebase.org/search/species/detail/?species_id=34779 about Trentepohlia jolithus? (The name is currently applied to Trentepohlia iolithus (Linnaeus) Wallroth and there is no reason to suppose that it is incorrect (Ross & Irvine, 1967). At least, follow always the same criteria: in Fig. 2 both names (T. iolithus and T. jolithus) are used).

Re: thank you for your comments. We agree with your opinion, and we have corrected in our new Figure 2.

3. Ln. 62-66 In my opinion, the objectives of the work should be presented clearly and in a more adequate format and without mentioning results. Something like: The aims of our study were: 1) to study the taxonomy of … 2) to present the complete chloroplast genome of … and 3) to reconstruct phylogenetic relationships ....

Re: thank you for your suggestion. We have rephrased this paragraph. “During 2010-2016, we collected lots of Trentepohliacean specimens from China reported in previous study [5]. In present study, we identified one corticolous specimen as T. odorata based on morphological evidence. Short- and long-read high-throughput sequencing data of this isolate were obtained and assembled. The aims of our study were: (1) to present the complete chloroplast genome of Trentepohlia odorata, (2) to study the taxonomy of Trentepohlia odorata, (3) to reconstruct phylogenetic relationships between Trentepohlia odorata and other species.”

4. Ln. 98 I think that the sentence ‘Maybe the main difference between two specimens was their localities and host trees.’ should be omitted. This reports to a difference in habitat and not to differences in specimen’s morphology.  Besides, this species is common in Fagus species.

Re: this sentence has been deleted in our revised manuscript.

5. Ln. 139 Taxonomic arrangement was following AlgaeBase’ – replace by ‘Taxonomic arrangement follows AlgaeBase’

Re: fixed.

6. Ln. 200-204  Please check the English in these lines. It is hard to understand.

Ln. 228-239  Please check English.

Re: thank you for your suggestion. We have asked a English native friend to help us revised these sentences.

7. Ln. 203 – ‘little pictures’ or ‘few images’?

Re: “few images”. Corrected.

8. Ln. 224 ‘could not’ or ‘cannot’?

Re: “cannot”, fixed.

9. Ln 225was not’ or ‘were not’; ‘in in’ (repeated)

Re: thank you for your comments. We have rephrased this sentence. “However, we cannot rule out that Trentepohliales have a closest relationship with Cladophorales since species in Cladophorales was not included in our phylogenomic analysis.”

10. Ln. 228-229 recovered Trentepohlia odorata is a separate species.’ Replace by ‘… recovered Trentepohlia odorata as a separate species.’

Re: fixed.

11. Ln. 237 Since Trentepohliles is ‘absolutely aerial’ – perhaps ‘exclusively aerial’ is more correct.

Re: fixed.

Ln 238 - Trentepohliceaen or Trentepohliacean?

Re: fixed, “Trentepohliacean”

12. Ln 239 I’m not understanding why Trentepohlia aurea, Trentepohlia bosseae and Cephaleuros are particularly relevant. Please explain.

Re: thank you for your comment. According to previous studies, there are six (or more) main genus-level clades in Trentepohliaceae, namely Cephaleuros, Stomatochroon, Trentepohlia aurea clade, Trentepohlia arborum-Printzina lagenifera clade, Trentepohlia iolithus-Trentepohlia bosseae-Phycopeltis clade and some clade formed by uncultured eukaryotic clones. Since T. aurea is the typical species of the typical genus Trentepohlia, its plastome is very important to understand this group. Cephaleuros and Stomatochroon are parasitic or endophytic mainly on leaves and it is not free-living in natural habitat, which is very interesting. Besides, according to our unpublished data, the difference of plastome between different Cephlaeuros species is very large. Since our phylogenetic analysis showed Trentepohlia odorata fall into Trentepohlia iolithus-Trentepohlia bosseae-Phycopeltis clade, the future study on plastome of this group could be more comparative with Trentepohlia odorata.